# Nitric Oxide Acts as an Inhibitor of Postharvest Senescence in Horticultural Products

**DOI:** 10.3390/ijms231911512

**Published:** 2022-09-29

**Authors:** Yongchao Zhu, Mei Du, Xianping Jiang, Miao Huang, Jin Zhao

**Affiliations:** 1Key Laboratory of Plant Resource Conservation and Germplasm Innovation in Mountainous Region (Ministry of Education), Guizhou University, Guiyang 550025, China; 2College of Agriculture, Guizhou University, Guiyang 550025, China

**Keywords:** nitric oxide, postharvest, senescence, molecular interaction

## Abstract

Horticultural products display fast senescence after harvest at ambient temperatures, resulting in decreased quality and shorter shelf life. As a gaseous signal molecule, nitric oxide (NO) has an important physiological effect on plants. Specifically, in the area of NO and its regulation of postharvest senescence, tremendous progress has been made. This review summarizes NO synthesis; the effect of NO in alleviating postharvest senescence; the mechanism of NO-alleviated senescence; and its interactions with other signaling molecules, such as ethylene (ETH), abscisic acid (ABA), melatonin (MT), hydrogen sulfide (H_2_S), hydrogen gas (H_2_), hydrogen peroxide (H_2_O_2_), and calcium ions (Ca^2+^). The aim of this review is to provide theoretical references for the application of NO in postharvest senescence in horticultural products.

## 1. Introduction

NO is a redox-active gaseous compound that regulates diverse physiological processes in plants. Numerous studies have demonstrated a regulatory role of NO in seed germination [1], adventitious root formation [2], fruit ripening [3], abiotic stress [4,5,6,7], and biotic stress [8,9].

Horticultural products undergo rapid senescence after harvest at ambient temperatures. Postharvest senescence is an important biological process for fresh horticultural products, accompanied by a series of materials and energy metabolism, including cell wall softening [10], chlorophyll degradation [11], new pigment (carotenoid, lutein, and flavonoid) synthesis [10,11,12], volatile accumulation [13], and the change in soluble substance content [12,13,14]. Senescent products are susceptible to fungal pathogens, which lead to decay and a decline in quality. According to statistics from 2017, the average postharvest loss rate is about 15% to 20% due to postharvest senescence, resulting in a large amount of postharvest loss of fresh horticultural products, which seriously affects the commodity value and economic income.

Currently, the role of NO in postharvest senescence has been widely reported in cut flowers [15,16], vegetables [17], and fruits [18,19]. Studies have shown that NO plays an essential role in preventing postharvest senescence. As a result, we systematically reviewed and summarized the production of NO and the regulatory role of NO during the postharvest senescence process in plants for a deeper understanding of the mechanisms of NO-alleviated postharvest senescence.

## 2. NO Production in Plants

In higher plants, NO is generated through two pathways: enzymatic and non-enzymatic reaction pathways. The enzymatic reduction pathway is catalyzed by NAD(P)H-dependent nitrate reductase (NR) in the presence of nitrite to form NO [20]. In the enzymatic oxidation pathway, NO is primarily formed by the catalysis of NO synthase-like (NOS) activity; it can also be formed by the oxidation of S-nitroso glutathione or polyamine metabolism, but this pathway of NO production has not been well elaborated in plant cells [21]. Apart from NR and NOS, the production of NO is also catalyzed by other enzymes. By way of example, xanthine oxidoreductase (XOR) can catalyze the reduction of nitrates and nitrites to NO [22]. NO can be generated either by non-enzymatic chemical reduction of nitrite at acidic pH or by light-driven reactions in the presence of carotenoids [23,24]. Furthermore, NO can also be a byproduct of denitrification and nitrification of NH_4_^+^ [25].

## 3. NO Delays Postharvest Senescence

Horticultural plants are prone to rapid senescence after postharvest storage at ambient temperature. Postharvest senescence is affected by several factors, such as temperature [26], light [27,28], and some plant growth regulators [29,30,31]. Multiple studies have shown that NO is an effective way to delay postharvest senescence. 

### 3.1. Exogenous NO Delays Postharvest Senescence

The effect of NO on alleviation of postharvest senescence can be demonstrated by the exogenous application of NO on postharvest plants. In exogenous NO treatment, three methods are available: fumigation, immersion, and spraying. Fumigation with direct NO gas delayed the senescence in postharvest mangoes and peaches [32,33]. The immersion of NO gas solution and NO donor sodium nitroprusside (SNP) or *S*-nitrosoglutathione (GSNO) solution also delays the postharvest senescence in some fruits by inhibiting ethylene production and reducing respiration rates [34,35]. Additionally, spraying NO donor GSNO solution is commonly used to extend the postharvest life of blueberries by improving their concentrations of ascorbic acid and glutathione [36]. The effects of exogenous NO on delaying postharvest senescence in horticultural products are listed in Table 1.

### 3.2. Endogenous NO Production during Postharvest Senescence Process

The effect of NO on postharvest senescence can also be revealed by endogenous NO production during postharvest senescence processes delayed by environmental factors and some chemical substances. For example, UV-B treatment can maintain decreased fruit firmness and delay postharvest senescence in mangoes by enhancing endogenous NO levels [43]. Endogenous NO production and NOS activity were induced by 1-methylcyclopropene (1-MCP) in the senescence process in cut roses [16]. Likewise, melatonin (MT) led to an increase in NO content through an increase in NOS activity and upregulation of *PcNOS* transcript levels, which subsequently delayed senescence in peaches [37]. However, in cold-stored peaches, abscisic acid (ABA) can induce endogenous NO synthesis via the NR pathway [44]. Similarly, NO production was also triggered by hydrogen gas (H_2_) by enhancing NR activity, which mitigated postharvest senescence in cut rose flowers [45].

## 4. The Mechanism of NO-Regulated Postharvest Senescence

NO delays postharvest senescence by regulating various metabolism pathways, including ethylene biosynthesis, respiratory metabolism, cell wall metabolism, reactive oxygen species (ROS) metabolism, and energy metabolism (Figure 1). Moreover, a set of *senescence-associated genes* (*SAGs*) that drive postharvest senescence are regulated by NO during postharvest senescence processes.

### 4.1. The Inhibition of Ethylene (ETH) Biosynthesis

It is well known that an increase in endogenous ETH is a sign of senescence. Thus, inhibiting endogenous ethylene production is considered a useful method to delay postharvest senescence. Exogenous NO can inhibit ETH production, which delays postharvest senescence of horticultural products, including mangoes [32,33], peaches [33], and cut rose flowers [16]. In addition, the inhibition of ETH biosynthesis-related enzymes 1-aminocyclopropane-1-carboxylic acid (ACC) oxidase (ACO) and ACC synthase (ACS) activity and their expression levels is associated with NO-induced decreases in the endogen ETH during the postharvest senescence process [32,34]. Therefore, the positive effect of NO on postharvest senescence is largely dependent on the inhibition of the ETH biosynthesis pathway.

### 4.2. The Decrease of Respiratory Metabolism

Climacteric transition is generally regarded as an important signal of the initiation of senescence in climacteric plants, which affects the storage life of postharvest plants. Reducing the respiratory rate can effectively delay postharvest senescence and prolong the shelf life of horticultural crops. A previous study showed that NO treatment restrained the increase in the respiration rate and extended the postharvest life of water bamboo shoots [17]. The application of 10 μL of NO gas fumigation significantly inhibited the respiratory rate of climacteric plums and peaches, thereby extending their shelf life [33,46]. Rather than suppressing the respiratory rate of climacteric fruits, NO was also shown to depress the respiratory rate of non-climacteric fruit. For example, the respiratory rate was significantly inhibited by NO treatment throughout the entire storage period of winter jujube fruit [47].

### 4.3. The Activation of Cell Wall Metabolism

Generally, senescent fruits exhibit the symptom of softening as a result of cell wall degradation. Several degrading enzymes, including polygalacturonase (PG) and pectin methylesterase (PME), are involved in the degradation of the cell wall [48]. Changes in cell wall metabolism-related enzyme activity are responsible for the decrease in firmness affected by a set of abiotic factors. The application of exogenous NO maintains the decrease in firmness and extends the postharvest life of blueberries [36]. Similarly, cornelian cherries treated with 500 μM of NO donor SNP exhibited higher firmness, possibly resulting from the lower activity of PE and PME, which degrade cell walls [49]. A decrease in the NO-induced activities of PG, PME, and *β*-galactosidase (*β*-Gal) delayed postharvest winter jujube fruit softening as well [50]. At the transcript level, NO treatment suppressed the softening of postharvest tomatoes by downregulating the gene expression levels of *LePG*, *LePhy1*, and *LePME* [34]. In summary, NO can inhibit cell wall metabolism-related enzyme activities, which maintains the decrease in firmness of horticultural products during the NO-delayed postharvest senescence process.

### 4.4. The Regulation of ROS Metabolism

Postharvest senescence is often accompanied by increased ROS, followed by the induction of some *SAGs*. The increase in ROS level occurs in parallel with increases in lipid peroxidation in senescent cells. In addition to endogenous ROS, ROS-related antioxidant enzymes are also related to postharvest senescence. Extensive research has shown that NO can delay postharvest senescence by decreasing ROS levels and enhancing antioxidant activities. Exogenous NO fumigation reduces ROS, O_2•_^−^_,_ and hydrogen peroxide (H_2_O_2_) contents but increases superoxide dismutase (SOD), peroxidase (POD), ascorbate peroxidase (APX), glutathione reductase (GR), and catalase (CAT) gene expression and enzymes activity, which prolongs the shelf life of table grape and appears to be strongly linked to the lipid peroxidation of membranes [51]. Likewise, NO enhances ROS scavenging capacity through the increased activity of SOD, CAT, APX, and GR, which are capable of diminishing the accumulation of O_2•_^−^ and H_2_O_2_, thereby delaying the senescence of winter jujube [47]. Therefore, NO can decrease the accumulation of ROS and enhance the antioxidant system, which leads to delayed senescence in postharvest horticultural crops.

### 4.5. The Promotion of Energy Metabolism

The lack of energy caused by the impaired respiratory chain and reduced ATP synthesis leads to cellular breakdown and dysfunction during the postharvest senescence stage [52]. The maintenance of cellular ATP and energy levels can thus maintain the normal physiological activities of the tissues, thereby postponing postharvest senescence and prolonging the shelf life of horticultural products. Several studies have established that NO-delayed postharvest senescence is ascribed to the optimization of energy metabolism. NO treatment, for example, delayed the softness and weight loss of water bamboo by maintaining the integrity of the mitochondrial ultrastructure and enhancing ATP levels [17]. Furthermore, NO donor SNP treatment enhanced ATP synthase activity, ATP synthase CF1 alpha subunit (AtpA) content, and *AtpA* expression levels in the postharvest freshness of cut lilies [53]. 

### 4.6. The Induction of SAGs

Various external and internal signals are likely to activate a set of *SAGs* that drive postharvest senescence. During postharvest senescence, *SAGs* can be induced by NO to delay senescence. These NO-induced *SAGs* include various transcription factors (*ERFs*) and some structural genes encoding enzymes related to cell wall metabolism, ethylene biosynthesis, and antioxidants. The *SAGs* regulated by NO during the postharvest senescence process are listed in Table 2.

## 5. Crosstalk between NO and Plant Growth Regulators during Postharvest Senescence

NO is generally accepted as a signaling molecule that alleviates postharvest plant senescence. Additionally, other plant growth regulators influence postharvest senescence, including ETH, ABA, MT, hydrogen sulfide (H_2_S), H_2_, H_2_O_2_, and calcium ions (Ca^2+^).

### 5.1. Crosstalk between NO and ETH

ETH, a gaseous plant hormone, is crucial for postharvest senescence. The inhibition of ETH action with NO has proven to be an excellent method for preventing postharvest senescence in cut flowers [16,59], fruits [18,38], and vegetables [34]. NO inhibits endogenous ETH production by reducing ACO activity during the postharvest senescence process in cut rose flowers and apple fruit [16,18]. Additionally, NO can suppress the synthesis of ETH and the expression of the ETH synthesis-related genes *ACO1* and *ACSs* in postharvest fruits [18,34]. Toxicological evidence has shown that ETH attenuates the delayed effect of NO during the postharvest senescence process. For example, the effect of NO on delaying postharvest senescence was improved by ETH inhibitor 1-MCP in tomatoes [60]. Moreover, it has also been suggested that 1-MCP delays cut rose flower senescence through the promotion of NOS activity and NO production [16]. Hence, NO and ETH repress each other by inhibiting ACO/ACS and NOS activity to regulate postharvest senescence in plants, respectively (Figure 2).

### 5.2. Crosstalk between NO and ABA

ABA is a vital phytohormone that regulates plant growth and development as well as abiotic and biotic stress. In addition, ABA is a regulator of postharvest senescence in cut rose flowers and leafy vegetables [61,62]. ABA increased NO content, NR activity, and *NR* transcript levels but inhibited NOS-like activity in peaches during cold storage [44]. However, exogenous NO did not affect ABA synthesis in postharvest peaches [44]. Nevertheless, NO biosynthesis inhibitor suppressed the effect of ABA on rose senescence [61]. Thus, NO might be involved in ABA-delayed postharvest senescence as a downstream signal molecule.

### 5.3. Crosstalk between NO and MT

A new and multifunctional hormone, MT, delays the postharvest senescence of many vegetables, fruits, and cut flowers, including broccoli [63], strawberries [64], and carnations [65]. Studies have shown that NO is involved in MT-delayed postharvest senescence. For example, exogenous MT enhances NOS and NR activity and promotes NO production, thereby suppressing postharvest pear senescence [37]. Moreover, the delaying effect of MT on fruit senescence was eliminated by N omega-nitro-L-arginine methyl ester (L-NAME), an inhibitor of NOS activity [37]. Therefore, the NO molecule appears to be downstream of MT during the postharvest senescence process.

### 5.4. Crosstalk between NO and H_2_

H_2_, a new signaling molecule, has been found to be involved in important physiological processes in plants, including germination [66], lateral and adventitious rooting [67,68], and plant tolerance against abiotic stress [69,70,71]. Recently, the roles of H_2_ in delaying postharvest senescence have been reported in cut flowers [72,73], fruits [74,75], and vegetables [76,77]. Similar to the role of H_2_ in adventitious root development [78] and stomatal closure [79], in which NO is involved, crosstalk between H_2_ and NO has also been reported during the postharvest senescence process. For example, the positive effect of H_2_ on alleviating the postharvest senescence of cut lilies was retarded by NO inhibitors [53]. Additionally, the generation of NO induced by H_2_ through the promotion of NR activity also delayed the postharvest senescence and prolonged the vase life of cut roses [45]. Thus, NO may be a downstream signaling molecule in H_2_-delayed postharvest senescence in plants (Figure 3).

### 5.5. Crosstalk between NO and H_2_S

The messenger molecule H_2_S is considered crucial to the process of postharvest senescence in plants [80,81,82]. Postharvest senescence is characterized by complex interactions between H_2_S and other messengers, such as NO. Recent studies suggest that H_2_S and NO have both synergistic and antagonistic effects on the postharvest senescence of plants. Cotreatment with H_2_S and NO delays postharvest senescence of strawberries and is better than H_2_S and NO treatments separately, suggesting a synergistic effect of H_2_S and NO [83]. Similarly, H_2_S has been shown to play a synergistic role with NO in delaying postharvest senescence in peach fruit [84]. Conversely, a study on NO delaying peach postharvest senescence through decreasing H_2_S content demonstrated an antagonistic relationship between H_2_S and NO [85]. Additionally, the H_2_S content of sweet pepper fruit increased, while the NO content decreased during the ripening process [86]. The interaction between H_2_S and NO differs between species, treatment methods, and developmental stages. However, further investigation into how H_2_S and NO interact during postharvest senescence is needed.

### 5.6. Crosstalk between NO and H_2_O_2_

H_2_O_2_ is generally regarded as an ROS that is a central regulator of plant physiological processes. H_2_O_2_ is also required for postharvest senescence in cut flowers [87]. The overproduction of endogenous H_2_O_2_ is not beneficial for delaying postharvest senescence in cut flowers [88]. Thus, inhibiting H_2_O_2_ accumulation is a useful method to delay postharvest senescence. Applying NO can prevent wound-induced browning and delay senescence by inhibiting the over-accumulation of H_2_O_2_ in fresh-cut lettuce [42]. In table grapes, NO activates catalase (CAT) activity and increases the transcript level of *CAT* to effectively decompose H_2_O_2_, which delays the postharvest senescence of fruit [51]. However, whether H_2_O_2_ delays postharvest senescence through the regulation of NO metabolism remains unclear. 

### 5.7. Crosstalk between NO and Ca^2+^

Ca^2+^ is a secondary messenger that plays a critical role in horticultural products’ senescence after harvest [89]. Several recent studies have indicated that Ca^2+^ interacts with NO during the postharvest senescence process. The application of the NO donor S-nitro-N-acetyl-penicillamine (SNAP) delays postharvest senescence and prolongs the vase life of cut lily flowers with increased Ca^2+^ and calmodulin (CaM) contents [15]. A Ca^2+^ chelator, Ca^2+^ channel inhibitor, and CaM antagonist reversed the NO-induced positive effect on cut lily flowers as well [15]. Accordingly, Ca^2+^/CaM may function as downstream molecules of NO during postharvest senescence.

To summarize, evidence has shown that NO can delay postharvest senescence and prolong shelf life by constructing an interacted network with other signaling molecules, including ethylene, ABA, H_2_S, MT, H_2_, H_2_O_2_, and Ca^2+^. Within this network, NO acts either as a downstream signaling molecule of some molecules (ABA, MT, and H_2_) by regulating NOS and NR activities and expression levels or as an upstream signal transducer of some molecules Ca^2+^/CaM and H_2_O_2_ (Figure 4). Nevertheless, NO and other signaling molecules (ETH and H_2_S) appear to cascade in a two-sided manner (Figure 4). Furthermore, ERFs might be involved in NO-regulated metabolism changes, including cell wall, respiratory, energy, and ROS metabolism, which delay postharvest senescence (Figure 4).

## 6. Conclusions and Outlook

In conclusion, NO generated by enzymatic (NOS and NR) and non-enzymatic reaction pathways is involved in delaying postharvest senescence in horticultural crops. Additionally, postharvest senescence can be delayed by the exogenous application of NO donors in cut flowers, vegetables, and fruits. A set of metabolism pathways, including ethylene biosynthesis, respiratory metabolism, cell wall metabolism, reactive oxygen species (ROS) metabolism, and energy metabolism, are regulated by NO, which is associated with postharvest senescence. Moreover, the crosstalk mechanisms between NO and other molecules (ETH, ABA, MT, H_2_S, H_2_, H_2_O_2_, and Ca^2+^) during the postharvest senescence process were reviewed. 

Although complex interactions between NO and other molecules have been observed, additional research is needed to determine how these signaling molecules influence NO biosynthesis metabolism and how NO regulates the metabolism of other molecules. Moreover, we need to discover whether other signaling molecules are interconnected in this network.

## Figures and Tables

**Figure 1 ijms-23-11512-f001:**
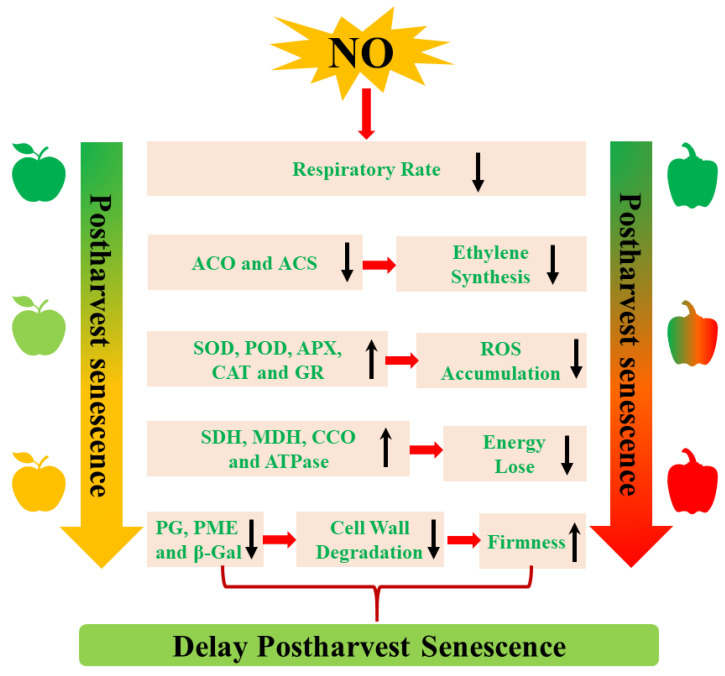
NO-regulated metabolism pathways during postharvest senescence. ACO, 1-aminocyclopropane-1-carboxylic acid oxidase; ACS, 1-aminocyclopropane-1-carboxylic acid synthase; APX, ascorbate peroxidase; CAT, catalase; CCO, cytochrome oxidase; ETH, Ethylene; GR, glutathione reductase; MDH, malic acid dehydrogenase; NO, nitric oxide; PG, polygalacturonase; PME, pectinmethylesterase; POD, peroxidase; ROS, reactive oxygen species; SDH, succinic dehydrogenase; SOD, superoxide dismutase; *β*-Gal, *β*-galactosidase. Upward arrow indicates up-regulation; Downward arrow indicates down-regulation.

**Figure 2 ijms-23-11512-f002:**
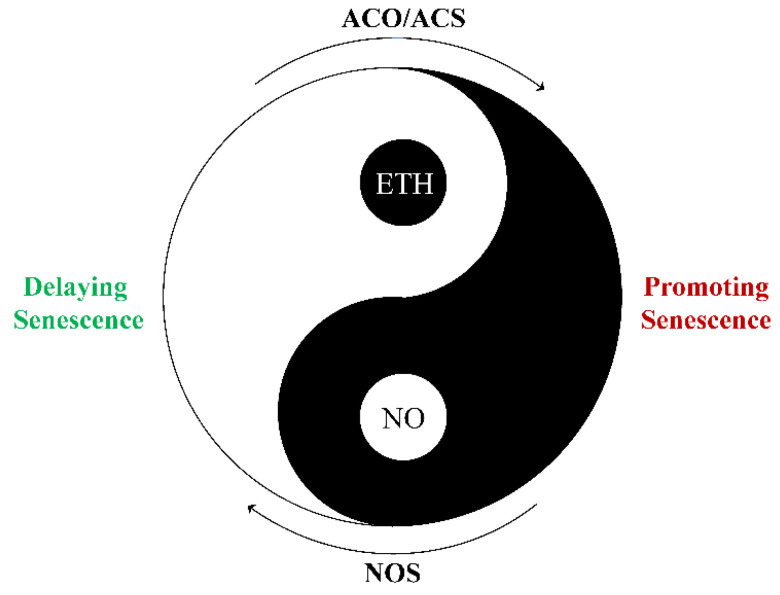
Model for the crosstalk between ETH and NO during the postharvest senescence process. The ‘Yin–Yang’ symbol represents the balance of ETH and NO generation through ACS/ACO and NOS pathways, respectively. ACO, 1-aminocyclopropane-1-carboxylic acid oxidase; ACS, 1-aminocyclopropane-1-carboxylic acid synthase; ETH, ethylene; NO, nitric oxide; NOS, NO synthase.

**Figure 3 ijms-23-11512-f003:**
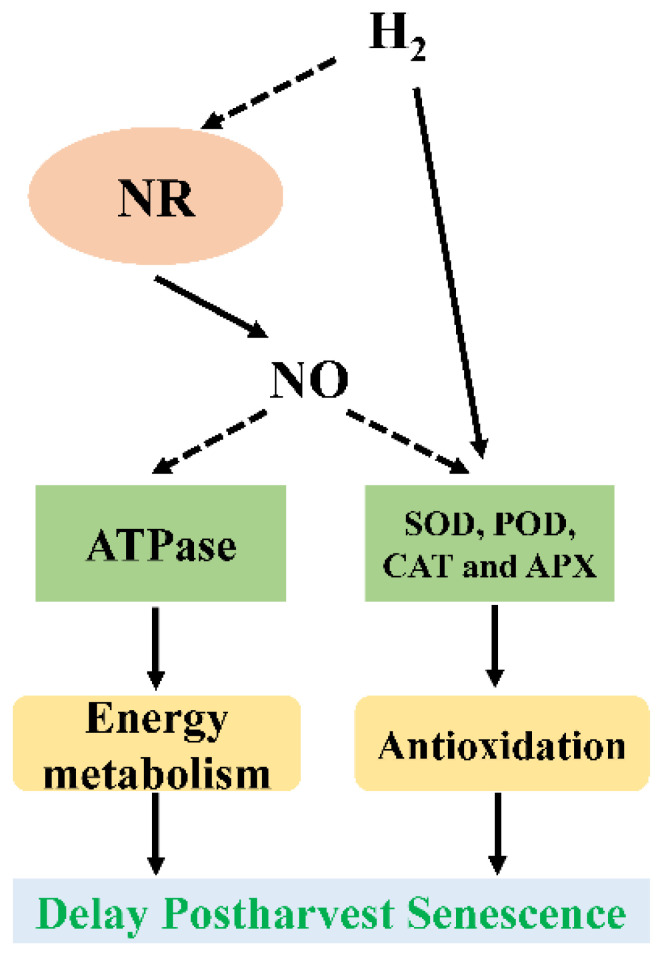
Schematic model depicting the requirement of NO in H_2_-delayed senescence of cut flowers. APX, ascorbate peroxidase; CAT, catalase; H_2_, hydrogen gas; NO, nitric oxide; NR, nitrate reductase; POD, peroxidase; SOD, superoxide dismutase.

**Figure 4 ijms-23-11512-f004:**
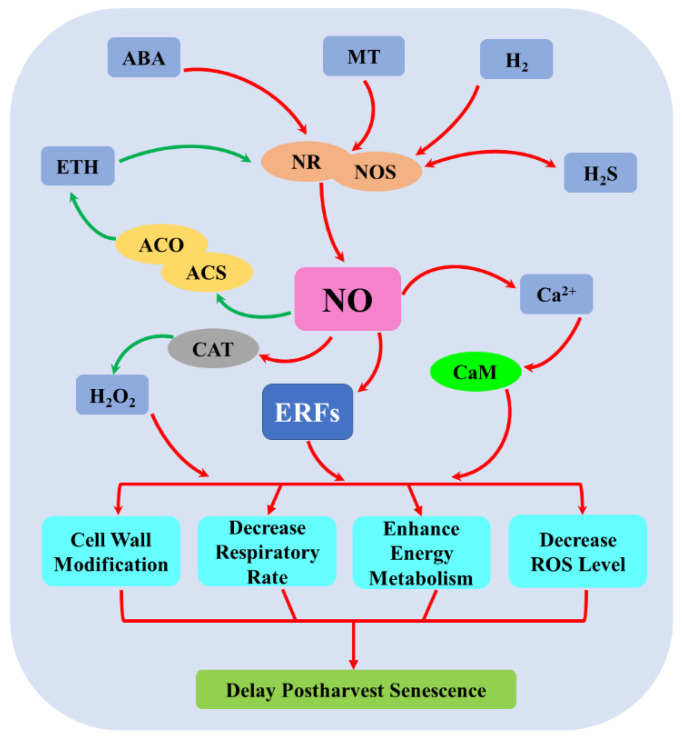
The crosstalk between NO and other molecules. ABA, abscisic acid; ACO, 1-aminocyclopropane-1-carboxylic acid oxidase; ACS, 1-aminocyclopropane-1-carboxylic acid synthase; Ca^2+^, calcium ion; CAT, catalase; CaM, calmodulin; ERFs, ethylene response factors; ETH, ethylene; H_2_O_2_, hydrogen peroxide; H_2_S, hydrogen sulfide; H_2_, hydrogen gas; MT, melatonin; NO, nitric oxide; NOS, nitric oxide synthase; NR, nitrate reductase. Red arrow indicates promotion; green arrow indicates inhibition; red two-way arrow indicates synergistic role; green two-way arrow indicates antagonistic role.

**Table 1 ijms-23-11512-t001:** Effects of NO on postharvest senescence in horticultural products.

Species	Treatment	NO-Mediated Effect	References
Pear	100 μM L^−1^ SNP	Decreased the transcript levels of cell wall- and ethylene synthetase-related genes; reduced respiration rate and ethylene production	[37]
Apple	100 μM L^−1^ GSNO	Activated nucleocytoplasmic MdERF5 and suppressed ethylene biosynthesis	[18]
Strawberry	5 μM L^−1^ SNP	Inhibited ethylene production, respiration rate, and activity of ACC synthase; reduced the content of ACC	[38]
Peach	10 μL L^−1^ NO	Maintained higher sucrose content but decreased glucose and fructose to lower levels during late storage	[33]
Carnation	0.1 mM L^−1^ SNP	Maintained water metabolism and antioxidative enzyme activity and mass-eliminated ROS as well as cell membrane stability	[39]
Rose	200 μM L^−1^ SNP	Decreased ethylene output by inhibiting ACO activity in cut rose flowers	[16]
Lily	100 μM L^−1^ SNAP	Increased Ca^2+^/CaM contents, enhanced Ca^2+^-ATPase activity, and up-regulated gene expression of *CaM*, *CBL1*, and *CBL3*	[15]
*Consolida ajacis* L.	40 μM L^−1^ SNP	Alleviated deteriorative postharvest changes by modulating physiological and biochemical mechanisms underlying senescence	[40]
*Calendula* *officinalis* L.	100 μM L^−1^ SNP	Improved flower longevity by delaying neck bending, inhibited bacterial growth, and increased activities of antioxidant enzymes	[41]
Tomato	1 mM L^−1^ SNP	Retarded pericarp reddening of tomato fruit, suppressed ethylene production, and influenced quality parameters during storage	[34]
Water bamboo shoots	30 μL L^−1^ NO	Delayed softness and weight loss and enhanced ATP levels by activating the expression and activity of SDH, MDH, and CCO	[17]
Lettuce	100 and 200 ppm NO	Inhibited the accumulation of H_2_O_2_, delayed senescence, and prolonged shelf life	[42]

**Table 2 ijms-23-11512-t002:** NO-regulated *SAGs* during postharvest senescence process.

Horticultural Products	Species	*SAGs*	References
Fruits	Pear	*PcPG*, *PcCel*, *PcACO1*, *PcACO2*, *PcACS1*, *PcNOS*, *PcNR1*, and *PcNR2*	[37]
	Apple	*MdACS1*, *MdACO1*, *MdERF5*, and *MdPP2C57*	[18]
	Mango	*MiACO*, *MiACS*, *MiETR1*, *MiERS1*, *MiEIN2*, and *MiERF*	[54]
	Table grape	*VvSOD*, *VvCAT*, *VvPOD2*, and *VvGR*	[51]
	Kiwifruit	*PG*, *PL*, *β-Gal*, *PE*, *ACO*, *ERS1*, *ETR2*, *ERF016*, *ERF7*, *ERF010*, *ERF062*, *ERF110*, *ERF037*, *ERF008*, *ERF113*, *ERF12*, *ERF095*, *CNGC1*, *CPK1*, *CIPK2*, *CML31*, *CML48*, and *ZIFL1*	[55]
	Wax apple	*PAL*, *POD*, *GLU*, *C3H*, *CA*, *F5H*, *4CL*, *CCoAOMT*, and *C4H*	[56]
	Peach	*PpaSOD*, *PpaCAT*, *PpaPOD*, *PpaPOD-1*, *PpaAPX*, and *PpaPAL*	[57]
Cut flowers	Gladiolus	*GgCyP1* and *GgDAD1*	[58]
	Lily	*CaM*, *CBL1, CBL3*, and *LlatpA*	[15,53]
Vegetables	Tomato	*LeACS2*, *LeACS4*, *LeACO1*, *LePME*, *LePG*, *LePhy1*, and *LeGAPDH*	[34]
	Water bamboo shoots	*ZlH^+^-ATPase*, *ZlNa^+^-K^+^-ATPase*, *ZlCa^2+^-ATPase*, *ZlMDH*, *ZlSDH*, and *ZlCCO*	[17]

4CL, 4-coumarate−CoA ligase; ACO, 1-aminocyclopropane-1-carboxylic acid oxidase; ACS, 1-aminocyclopropane-1-carboxylic acid synthase; APX, ascorbate peroxidase; AtpA, ATP synthase CF1 alpha subunit; C3H, *p*-coumarate 3-hydroxylase; C4H, *trans*-cinnamate 4-monooxygenase; CA, coniferyl-aldehyde dehydrogenase; CaM, calmodulin; CAT, catalase; CBL, calcineurin B-like protein; CCO, cytochrome oxidase; CCoAOMT, caffeoyl-CoA O-methyltransferase; Cel, cellulase; CIPK, calcineurin B-like protein-interacting protein kinase; CML, calmodulin-like protein; CNGC, cyclic nucleotide-gated channel; CPK, calcium-dependent protein kinase; CyP, cysteine protease; DAD, defender against death; EIN, ethylene insensitive; ERF, ethylene response factor; ERS, ethylene response sensor; ERT, ethylene receptor; F5H, ferulate-5-hydroxylase; GAPDH, glyceraldehyde 3-phosphate dehydrogenase; GLU, *β*-glucosidase; GR, glutathione reductase; MDH, malic acid dehydrogenase; NOS, nitric oxide synthase; NR, nitrate reductase; PAL, phenylalanine ammonia-lyase; PE, pectin esterase; PG, polygalacturonase; Phy, phytoene synthase; PL, pectate lyase; PME, pectin methylesterase; POD, peroxidase; SAGs, senescence associated genes; SDH, succinic dehydrogenase; SOD, superoxide dismutase; ZIFL, calmodulin-binding heat-shock protein; *β*-Gal, *β*-galactosidase.

## Data Availability

Not applicable.

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
