# Peer review of "Nitric Oxide Acts as an Inhibitor of Postharvest Senescence in Horticultural Products"

_ijms, 2022, doi:10.3390/ijms231911512_

Round 1
Reviewer 1 Report
The authors presented that NO is a redox-active gaseous compound that have many role in plant development, seed germination etc.
They presented how NO is produced in plants, how it may delay the postharvest senescence in a exogenous application and they give some plant examples as well as the concentration that was used and the endogenous production. Then the authors present the regulation of NO in postharvest dividing in subtopics. The next topic is the crosstalk with NO and plant hormones. The review is really interesting but some topic may be better explored giving more information in special to enzymes and genes regulation.
I would suggest:
1) To improve the topic about the endogenous NO production, presenting in more detail the hormones roles and croostalk and genes envolved
2) The ROS regulation can be explain better as well the cell wall as these topics are really important to plant and it may connected to others abiotic stress
3) It would present a final model to connect all the information presented.
Reviewer 2 Report
Useful review for the complex problem - regulation of postharvest senedcence.
Reviewer 3 Report
This review article entitled “Nitric Oxide Acts as An Inhibitor of Postharvest Senescence in Horticultural Products” by Zhu et. al. summarizes NO synthesis, the effect of NO in alleviating postharvest senescence, the mechanism of NO-alleviated senescence and its interactions with other signaling molecules, such as ethylene (ETH), abscisic acid (ABA), melatonin (MT), hydrogen sulfide (H2S), hydrogen gas (H2), hydrogen peroxide (H2O2), calcium ion (Ca2+). This review provides theoretical references for the application of NO in postharvest senescence of horticultural products. In my view, this review is logically planned and executed, and recent findings are appropriately interpreted. However, I have some concerns and amendments that need to be thoroughly addressed prior to the final publication.
General Comments:
Please careful check the English language. Some sentences are not well structured and several grammatical errors in the manuscript.
Introduction:
Line 28: Please provide some statistical data about loss of horticultural crops due to postharvest senescence.
Line 270: Please shift figure 4 from the conclusion. “Conclusions and Outlook” should be the summary and research need in this area.
